# A Stellar magnesium to silicon ratio in the atmosphere of an exoplanet

Jorge A. Sanchez [1] ✉, Peter C. B. Smith[1], Krishna Kanumalla[1], Luis Welbanks [1], Michael R. Line [1,16] ✉, Stefan Pelletier [2], Steven Desch[1], Patrick Young [1], Jennifer Patience[1], Jacob Bean[3], Matteo Brogi[4,5], Dan Jaffe[6], Gregory N. Mace [6], Megan Weiner Mansfield [7], Vatsal Panwar[8,9,10], Vivien Parmentier [11], Lorenzo Pino [12], Arjun Baliga Savel [7], Lennart van Sluijs[13] & Joost P. Wardenier[14,15]

The elemental compositions of exoplanets encode information about their formation environments and internal structures. While volatile ratios such as carbon-to-oxygen (C/O) are used to trace formation location, the rock-forming elements–magnesium (Mg), silicon (Si), and iron (Fe)–govern interior mineralogy and are commonly assumed to reflect the host star's abundances. Yet this assumption remains largely untested. Ultra-hot Jupiters, gas-giant exoplanets with dayside temperatures above 3000 K, provide rare access to refractory elements that remain gaseous. Here we present high-resolution thermal emission spectroscopy of the exoplanet WASP-189b ($T_{eq} = 3354^{+27}_{-34}$ K) obtained with the Immersion Grating Infrared Spectrometer (IGRINS) on Gemini South. We detect neutral iron (Fe I), magnesium (Mg I), silicon (Si I), water ($H_2O$), carbon monoxide (CO), and hydroxyl (OH) at signal-to-noise ratios exceeding 4, and retrieve their elemental abundances. We show that the Mg/Si, Fe/Mg, and Si/Fe ratios are consistent with stellar values, while the refractory-to-volatile ratio is enhanced by roughly a factor of 2. These findings demonstrate that giant-planet atmospheres can preserve stellar-like rock-forming ratios, providing an empirical validation of the stellar-proxy assumption that underpins planetary composition and formation models across exoplanet systems.

Measurements of the elemental composition of planets enable fundamental constraints on how planets form, migrate, and evolve[1,2]. Exoplanet atmospheres of giant planets, in particular, offer a unique window[3] into the processes occurring within proto-planetary disks that shape planetary systems, augmenting stellar and Solar-System material abundance measurements. Volatile (ices), elements such as carbon and oxygen have been extensively used to trace the origins and migration histories of giant planets[4–7], revealing a wide diversity in

[1]School of Earth and Space Exploration, Arizona State University, Tempe, AZ, USA. [2]Observatoire astronomique de l'Université de Genève, Versoix, Switzerland. [3]Department of Astronomy and Astrophysics, University of Chicago, Chicago, IL, USA. [4]Dipartimento di Fisica, Università degli Studi di Torino, Torino, Italy. [5]INAF – Osservatorio Astrofisico di Torino, Pino Torinese, Italy. [6]Department of Astronomy, The University of Texas at Austin, Austin, TX, USA. [7]Department of Astronomy, University of Maryland, College Park, MD, USA. [8]Department of Physics, University of Warwick, Coventry, UK. [9]Center for Exoplanets and Habitability, University of Warwick, Coventry, UK. [10]School of Physics and Astronomy, University of Birmingham, Birmingham, UK. [11]Laboratoire Lagrange, Observatoire de la Côte d'Azur, CNRS, Université Côte d'Azur, Nice, France. [12]INAF – Osservatorio Astrofisico di Arcetri, Florence, Italy. [13]Department of Astronomy, University of Michigan, Ann Arbor, MI, USA. [14]Trottier Institute for Research on Exoplanets (iREx), Université de Montréal, Montréal, QC, Canada. [15]Physics Institute, Space Research and Planetary Sciences, University of Bern, Bern, Switzerland. [16]These authors jointly supervised this work: Michael R. Line. ✉e-mail: jasanchez@asu.edu; mrline@asu.edu

potential formation scenarios[8] across the exoplanet population. However, these volatile diagnostics are strongly influenced by condensation fronts (ice-lines) and other disk chemical processes[9] limiting their utility in fully linking present-day atmospheric composition to the processes that sculpted their birth[7].

References[10,11] highlighted the utility of giant planet atmosphere refractory elemental (e.g., Fe, Mg, Si) constraints as a tracer of the rocky material within protoplanetary disks. Because these elements condense at high temperatures, their relative abundances are expected to remain nearly constant throughout the disk[11,12], providing a compositional baseline that links stars, giant planets, and terrestrial bodies. Measurements of volatile-to-refractory enrichments in ultra-hot Jupiters (UHJs)–where both volatile and refractory species persist in the gas phase within the atmosphere–have begun to reveal how rocks and ices are incorporated into planetary envelopes[10,11,13,14]. Furthermore, the same rock-forming ratios (Mg/Si and Fe/Mg) govern the mineralogy, core size, and mantle rheology of terrestrial planets[15–17]. For these worlds, direct elemental composition measurements are not yet possible; hence, modeling investigations of terrestrial exoplanets inherently assume that the bulk planetary composition reflects that of the host star. Testing this assumption observationally by measuring refractory ratios in giant-planet atmospheres provides the necessary benchmark for interpreting the composition of rocky and giant planets alike.

Here we show, with high-resolution thermal emission spectroscopy of the ultra-hot Jupiter WASP-189b, direct constraints on both refractory (Mg, Si, Fe) and volatile (C, O) elemental ratios. We find that WASP-189b's atmospheric refractory elemental ratios reflect those of its host star, providing an empirical validation of the assumption of using stellar abundances as a proxy for refractory composition.

## Results
### Observations
We obtained high-resolution ($R$ approximately 45,000) thermal emission spectra of WASP-189b (planet radius, $R_P = 1.6 \pm 0.017\ R_{Jup}$, planet mass, $M_P = 1.99 \pm 0.16\ M_{Jup}$)[18,19] using IGRINS[20], formerly at Gemini South. Previous studies using IGRINS have detected individual volatile species ($H_2O$, CO, OH[21,22]) and the combination of multiple refractory species (Fe I, Mg I, Si I, Ti I, Ca I, Cr I, V I,[14]) in the atmospheres of giant planets, due to the instruments broad and near continuous wavelength coverage (1.4 - 2.5 $\mu$m) over which many of these species have spectral features (Fig. 1, panel b). Observations of WASP-189b were conducted over two separate nights (2022-05-07 and 2023-04-02 UTC), taken just prior to (night 1) and following (night 2) secondary eclipse, capturing the thermal emission of the planet's day-side hemisphere. Supplementary Fig. 1 shows the median SNR and humidity during the night for each observation. Over the course of each observation, dozens of individual spectra are taken at different phases in the planetary orbit

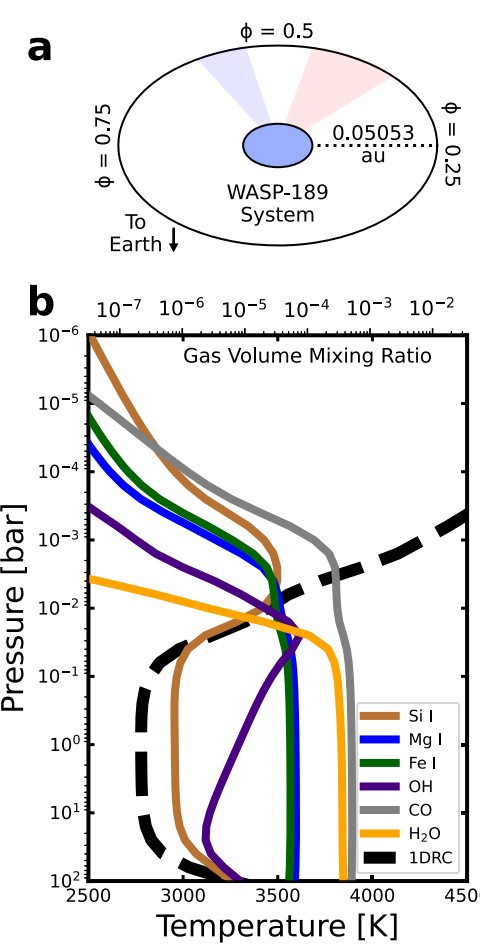
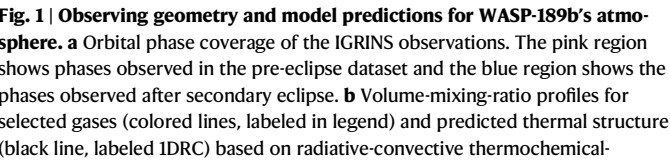
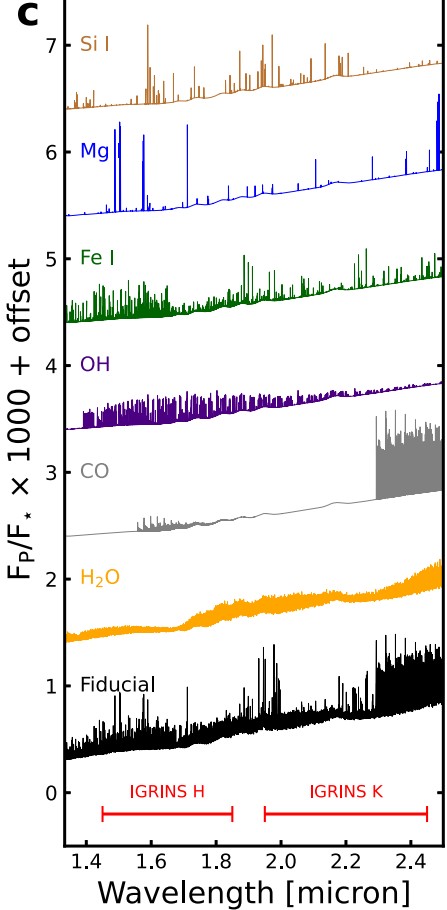

**Fig. 1 | Observing geometry and model predictions for WASP-189b's atmosphere. a** Orbital phase coverage of the IGRINS observations. The pink region shows phases observed in the pre-eclipse dataset and the blue region shows the phases observed after secondary eclipse. **b** Volume-mixing-ratio profiles for selected gases (colored lines, labeled in legend) and predicted thermal structure (black line, labeled 1DRC) based on radiative-convective thermochemical-equilibrium. **c** Model planet-to-star flux ratios illustrating the sensitivity of the IGRINS wavelength range to individual volatile and refractory species. The fiducial spectrum includes all species used in the cross-correlation analysis, and individual model spectra highlight key contributors across the bandpass. Source data are provided as a Source Data file.

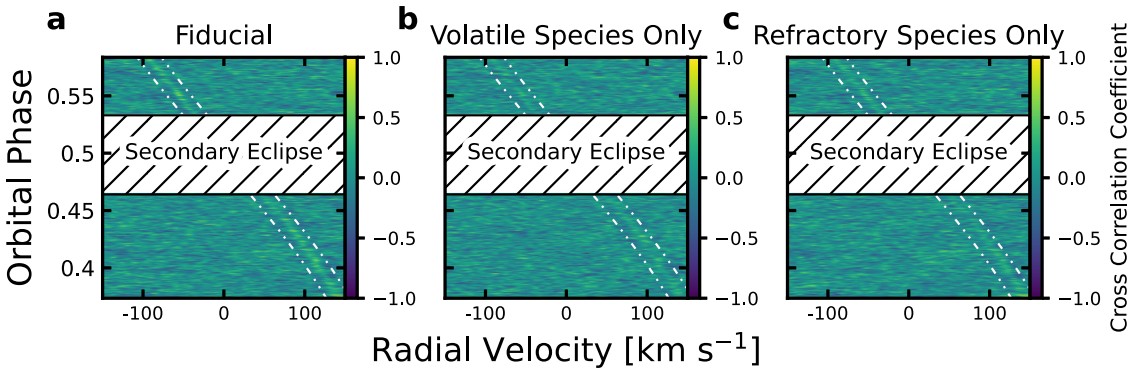

**Fig. 2 | Cross-correlation coefficient as a function of velocity and orbital phase arising from varying atmospheric model templates. a** Fiducial model, including both refractory (Fe I+, Mg I, Si I, Ti I, Ca I, V I) and volatile (H₂O, CO and OH) species. **b** Model including only volatile species. **c** Model including only refractory species. The colored trails indicate peaks of the cross-correlation function across the orbital phases covered by our observations, indicating a detection of atmospheric emission given that template. The white dashed lines denote ± 15 km/s offsets from the best fit velocity parameters measured in our atmospheric retrieval analysis. The white box in the middle of each panel indicate the phases during which the planet is blocked by the host star during secondary eclipse.

(Fig. 1, panel a). To isolate the faint planetary signal from Earth's telluric lines and the stellar features in each of our spectra, we apply standard detrending procedures following[21,23,24] (see Methods, subsection Observations and Data Reduction). The detrending process removes these dominant features, while preserving the planet signal within the residual data.

## Gas Detections

We generate a model spectrum (labeled Fiducial Model in Fig. 2), meant to match the thermal emission signal from the planet to compare with our post-detrended data. During our observations, the spectral lines of the planet are constantly Doppler-shifted by the planet's orbital motion. We therefore also shift the generated model spectrum along a range of line of sight velocities, and calculate a cross-correlation coefficient between the data and shifted model[14,25] (see Methods, subsection Gas Detection via Cross-Correlation). The peak value of the cross-correlation coefficient (CCF), will occur at the velocity shifts that best match the model with the observed data for that orbital phase. Repeating this process in time with all spectra taken throughout the night reveals a coherent trail in velocity-phase space that traces the planet's motion[21,23]. Figure 2 shows this trail for three model configurations: The first is our fiducial model (Fig. 2a), which is a combination of both volatile (H₂O, CO, OH) and refractory gases (Fe I, Mg I, Si I, Ti I, Ca I, V I). We included these gases in our fiducial model due to their previous detections on other ultra-hot Jupiters[22,26,27], and their expected volume mixing ratios of > 10⁻⁵ based on the theoretical gas abundance profiles for WASP-189b as shown in panel c of Fig. 1. Fig. 2b, c isolate the contribution of the volatile and refractory gases, using models containing each set of species separately. The clear presence of trails in all three configurations indicates that multiple species from both volatile and refractory groups are present in WASP-189b's dayside atmosphere.

To detect the signature of individual molecular and atomic species in the observed spectrum, we sum the correlation coefficient along the orbital path taken by the planet during our observations. By calculating this summed correlation coefficient as a function of different velocities (Keplerian velocity $K_p$, and an offset from the star-planet system velocity $dV_{sys}$ in Fig. 3), we isolate the total atmospheric signal for a given single model template. A species is considered detected if the peak signal–which should near the expected $K_p$ and $dV_{sys}$ pair–is detected at a signal-to-noise ratio (S/N) of at least 4 ("4σ") relative to the background[21] (see Methods, subsection Gas Detection via Cross-correlation). Figure 3 summarizes the detections of individual gases. We detect the following refractory and volatile species: Fe I (8.51σ), CO (6.22σ), Si I (5.91σ), OH (5.79σ), Mg I (4.71σ) and H₂O (4.36σ).

These detections add to a growing body of literature of WASP-189b observed at high spectral resolution. Fe I and Mg I have been previously detected on WASP-189b (along with numerous other atomic species) in both the optical[28,29] and in the near-infrared[30–32]. This work marks the only detection of water and significant (S/N > 5σ) detections of Si I and OH in the atmosphere of this planet.

## Elemental abundance determinations

To obtain quantitative estimates on the atmospheric elemental abundances, we employ Bayesian inference methods for parameter estimation, also known as retrievals[21,22,33,34]. Our framework couples an atmospheric model[21,22,25] with the affine-invariant ensemble Markov-chain Monte Carlo sampler[35], to obtain constraints on elemental abundances, vertical temperature structure, and the Keplerian/system velocities. The atmospheric forward model assumes[13,31] thermochemical equilibrium[36] for the molecular and atomic gases in WASP-189b given the elemental abundance ratios and the temperature-pressure profile. This parameterization is preferred for ultra-hot Jupiters as the high temperatures result in kinetic timescales that are short compared to disequilibrium processes (photochemistry, transport) timescales, resulting in equilibrium gas abundances throughout the atmosphere[37–39]. The elemental abundance parameters are of the form

$$[X/Y]_{\odot/*} = \log_{10}\left(\frac{(X/Y)}{(X/Y)_{\odot/*}}\right). \tag{1}$$

where $[X/Y]_{\odot/*}$ denotes $\log_{10}$ of species X relative to Y, relative to X/Y in the sun ($\odot$)[40]/star (*). Our model also includes a parameterized temperature-pressure profile prescription similar to one described in ref. 14 (see Methods, subsection Atmospheric Modeling and Retrieval Frameworks). All the parameters included in the model are listed with their sampled prior ranges in Table 1.

The results from our main retrieval are summarized in Supplementary Figs. 3–5. A full corner plot showing all 22 parameters is also available in the zenodo repository given in the Data Availability section. We compare the retrieved elemental abundances found in WASP-189b with those measured in the host star from[18]. These results are summarized in panel a of Fig. 4. The retrieved refractory abundances are consistent to within approximately 1σ to those measured in the star from ref. 18, while the C and O abundances are slightly sub-stellar (consistent within 2σ). We also compare our retrieved planet-to-star abundance ratios with other sets of stellar abundances of the host star found in the literature[41], summarized in Supplementary Table 1. Given the high temperatures of the atmosphere by WASP-189b, we note that we do not expect the depletion

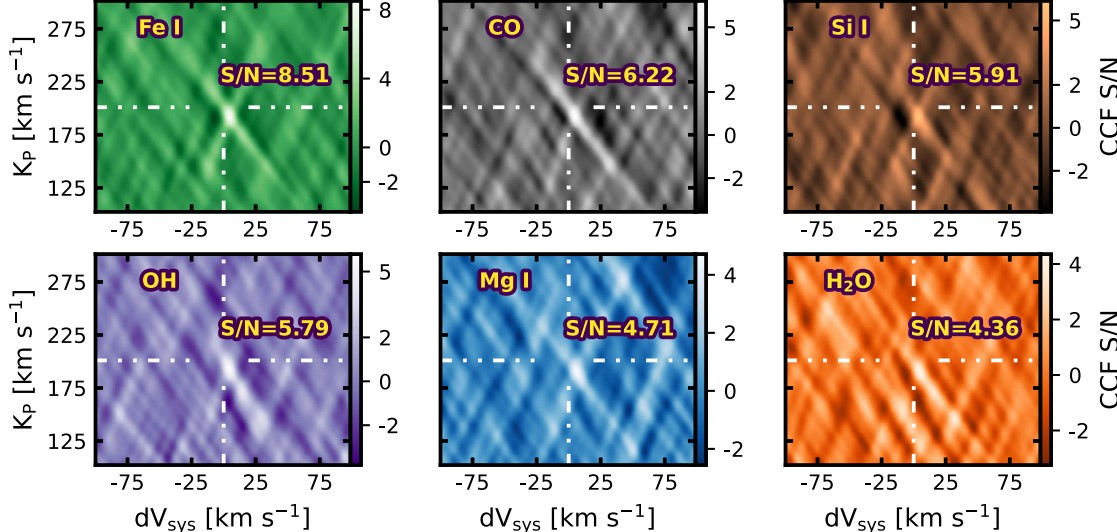

**Fig. 3 | Cross-correlation signal-to-noise (S/N) ratio maps illustrating the detection of individual species in atmosphere of WASP-189b.** The detected gas for each map is indicated in the upper left of the panel. If that gas is present, a peak occurs near the expected values for the planet's radial velocity semi-amplitude ($K_P$) and the offset from the star-planet system velocity ($dV_{sys}$)–indicated by the white dot-dashed lines. The S/N for each detection is indicated in each panel.

## Table 1 | Description of the retrieved parameters and their prior ranges

| Parameter | Description | Prior range |
|---|---|---|
| [O/H] | Oxygen enrichment relative to solar | $\mathcal{U}(-5, 5)$ |
| [C/H] | Carbon enrichment relative to solar | $\mathcal{U}(-5, 5)$ |
| [Ca/H] | Calcium enrichment relative to solar | $\mathcal{U}(-5, 5)$ |
| [Fe/H] | Iron enrichment to relative solar | $\mathcal{U}(-5, 5)$ |
| [Si/H] | Silicon enrichment relative to solar | $\mathcal{U}(-5, 5)$ |
| [Ti/H] | Titanium enrichment relative to solar | $\mathcal{U}(-5, 5)$ |
| [Mg/H] | Magnesium enrichment relative to solar | $\mathcal{U}(-5, 5)$ |
| [V/H] | Vanadium enrichment relative to solar | $\mathcal{U}(-5, 5)$ |
| T0 | Temperature, Node 1 [K] | $\mathcal{U}(100, 6000)$ |
| T1 | Temperature, Node 2 [K] | $\mathcal{U}(100, 6000)$ |
| T2 | Temperature, Node 3 [K] | $\mathcal{U}(100, 6000)$ |
| T3 | Temperature, Node 4 [K] | $\mathcal{U}(100, 6000)$ |
| T4 | Temperature, Node 5 [K] | $\mathcal{U}(100, 6000)$ |
| T5 | Temperature, Node 6 [K] | $\mathcal{U}(100, 6000)$ |
| $\log_{10}P_2$ | Pressure at Node 2 [bar] | $\mathcal{U}(-6, 2.5)$ |
| $\log_{10}P_3$ | Pressure at Node 3 [bar] | $\mathcal{U}(-6, 2.5)$ |
| $\log_{10}P_4$ | Pressure at Node 4 [bar] | $\mathcal{U}(-6, 2.5)$ |
| $\log_{10}P_5$ | Pressure at Node 5 [bar] | $\mathcal{U}(-6, 2.5)$ |
| $dKp_{1,2}$ | Difference in Planetary Velocity [km/s] | $\mathcal{U}(-20, 20)$ |
| $dVsys_{1,2}$ | Difference in Systemic Velocity [km/s] | $\mathcal{U}(-20, 20)$ |

The first column gives the title of each parameter in our main retrieval algorithm. Column 2 is the description of each parameter along with the associated units. Column 3 gives the prior ranges for each of these parameters.

of any elemental abundances due to night-side cold-trapping[42] (see Supplementary Fig. 8). Additionally, while many chemical species are expected to ionize at low pressures in UHJs[29,42], our use of a chemical equilibrium prescription for the atmosphere accounts for these processes. We therefore do not expect either process to impact our final abundance calculations.

From the retrieved elemental abundances, we compute $[Mg/Si]_\star = -0.18^{+0.20}_{-0.21}$, $[Mg/Fe]_\star = 0.05^{+0.22}_{-0.22}$ and $[Si/Fe]_\star = 0.24^{+0.18}_{-0.18}$. In line with expectations from similar analyses on other UHJs[13,14,29,42], the Mg/Fe and Si/Fe ratios are consistent with their reported stellar values (here relative to ref. [18]) at the 1 to 2 $\sigma$ confidence level. Our analysis also confirms this lack of divergence from the stellar composition for the Mg/Si ratio in WASP-189b, further supporting the assumption that all three species (Mg, Fe, Si) would remain in similar proportions to the stellar value within the disk[12,43]. Supplementary Fig. 7 also shows the consistency of our derived Mg/Si ratio against different listed stellar abundances for WASP-189[41] found in the literature as well as comparisons with the solar value and local FGK star Mg/Si ratio measurements.

We determine the total atmospheric metal (M—anything heavier than hydrogen, H, and helium, He, relative to H) enrichment and C/O ratio to be consistent with stellar values at the 68% confidence level ($[M/H] = -0.25^{+0.33}_{-0.26}$, C/O = $0.49^{+0.10}_{-0.09}$, the stellar C/O is 0.40[18]). We also measure a moderately super-stellar refractory (R=Fe+Mg+Si+Ti+ Ca +V)-to-volatile (V=C+O) ratio of $[R/V]_\star = 0.38^{+0.18}_{-0.19}$ ($2.42^{+1.21}_{-0.87}$ × the stellar value), and a refractory content [R/H]$_\star$ of $0.01^{+0.36}_{-0.29}$ × the stellar value. (See Supplementary Discussion).

## Discussion

As there are no other exoplanets with measured Mg/Si ratios, we weigh our results against other astrophysical objects with this measurement in panel b of Fig. 4. These include the Earth and Sun, CI carbonaceous and enstatite chondrites, local FGK stars (green)[44], and a subset of polluted white dwarfs (yellow)[45]. The spectra of polluted white-dwarfs are thought to encode the record of in-falling planetary or cometary material post main-sequence[46–48]. However, the elemental abundances of the progenitor star remain unknown due to gravitational settling, making it impossible (unless they exist within a binary star system[49]) to link the planetary and stellar composition for those systems. The measurement of the Mg/Si ratio on WASP-189b (red) allows for the direct comparison of a planets Mg/Si ratio with not only its host star, but also with other objects having this measured quantity, showing consistency with many polluted WDs, along with the solar value and local FGK population. Our result expands the diversity of environments in which the Mg/Si ratio has been measured, extending this key

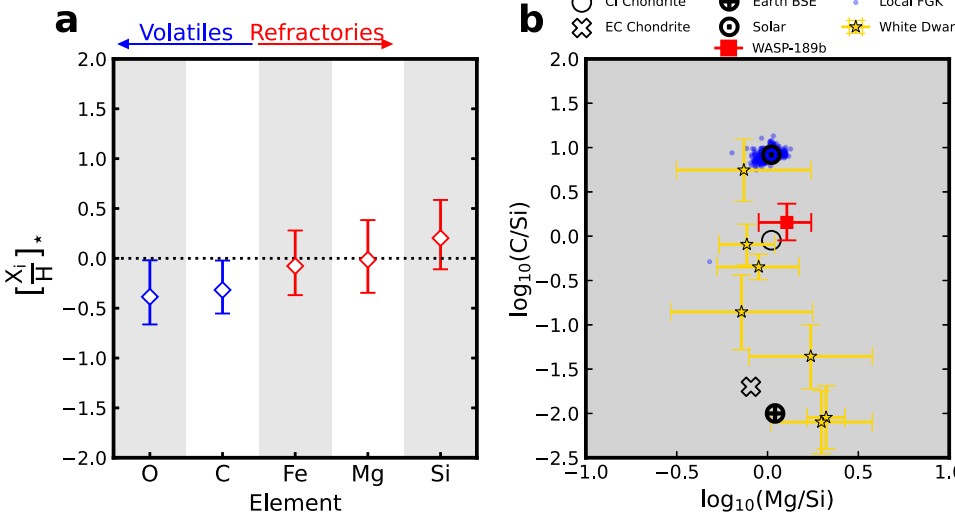

**Fig. 4 | Elemental abundance estimates and abundance ratios measured in WASP-189b. a** Abundance estimates for O, C, Fe, Mg, and Si in WASP-189b relative to those measured in the star, $[X_i/H]_*$,[18]. The errorbar represents the 68 % confidence interval for each measurement. The abundances of O and C are slightly substellar (by just over $1\sigma$), while the refractory species are stellar within $1\sigma$ uncertainties. **b** Logarithm of the ratio of carbon to silicon measured in different astrophysical objects as a function of the logarithm of the measured magnesium to silicon ratio. The values shown are for the Sun, Bulk Silicate Earth (BSE), CI and EC Chondrites, polluted White Dwarfs (yellow stars), local FGK stars (blue dots), and our measurement of WASP-189b (red square). Error bars represent the 68 % confidence interval on measurement when shown. The carbon to silicon values for non-stellar objects are from ref. 69. The FGK star values are taken from ref. 44. Figure adapted from ref. 45. The solar value for Mg/Si is assumed as 1.05, while the value for Earth BSE is assumed as 1.10[44,70]. Source data are provided as a Source Data file.

geochemical metric from Solar System materials and disrupted planetary debris to the atmosphere of a gas giant exoplanet.

To explore the geochemical context of our measured abundance ratios, we calculate the relative proportions of Mg, Si, and Fe in WASP-189b's atmosphere. Based on the retrieved abundances, we derive a Mg:Si:Fe ratio of approximately 1.2:1.0:0.7, which falls between the typical values found in enstatite and ordinary chondrites[50]. If a rocky planet were to form from material with this composition, its upper mantle would likely consist of a mixture of olivine and pyroxene, similar to Earth's[15,16]. While we do not claim that giant planet atmospheres and rocky planet interiors evolve identically, both draw from the same disk reservoir of refractory solids[12]. The atmospheres of gas giants–particularly those with minimal condensation or cold-trapping (like WASP-189b, see supplementary material, section Thermal Structure)–are expected to retain the elemental ratios of the accreted refractory species, thereby serving as observable tracers of the disk's bulk refractory composition[10,11]. The consistency of WASP-189b's atmospheric Mg/Si, Mg/Fe, and Si/Fe ratios with the host star measurements supports the assumption–widely used in terrestrial planet interior modeling[15,16]–that stellar refractory abundances trace the bulk composition of solids in the protoplanetary disk[11,12,17,51]. In this sense, measurements of these ratios in ultra-hot Jupiters offer an observational anchor for understanding the link between planet and host star composition, and ultimately the refractory building blocks in planetary systems.

By measuring the abundances of Si, Mg, and Fe in an exoplanet atmosphere, we show that the full set of rock-forming elemental ratios (Mg/Si, Mg/Fe, Si/Fe) exists in the same relative proportion as the host star, within measurement uncertainties. In doing so, we provide initial observational evidence that an Mg/Si ratio of an exoplanet matches that of its host-star, strengthening the stellar abundance ratio assumption used in interior models of rocky and volatile-rich exoplanets[43,51]. Along with measurements of volatile species such as carbon and oxygen, this multi-species analysis illustrates the wealth of information that can be gained from studying giant planets at high spectral resolution. Multi-wavelength, high-spectral resolution campaigns to study these kinds of systems both now and with the

upcoming Extremely Large Telescope's offer the opportunity to reveal the larger chemical inventory that exists within alien worlds. This effort will lead to even more precise measurements on the planet-star abundance patterns, for refined estimates on the chemical evolution and compositional diversity that exists within exoplanetary systems.

## Methods
### Observations and Data Reduction
Using the IGRINS instrument on (formerly on Gemini South[20,52]) we observed two separate half-nights of data capturing the direct thermal emission of the ultra-hot Jupiter WASP-189b. The first, taken on UTC 2022-05-07 as part of the Large and Long Program GS-2021B-Q-113 (PI M. Line–The Roasting Marshmallows Survey), consisted of a 4.82 hour long continuous sequence of 28s exposures in an AB-BA nodding pattern while the planet was in the pre-secondary eclipse phases ($0.373 < \phi < 0.464$). This pre-eclipse sequence had a total of 155 AB pairs (hereafter referred to as frames). The second sequence was taken on UTC 2023-04-02 as part of the Queue program GS-2023A-Q-231 (PI J. Sanchez) and consisted of a 3.23 hour long continuous sequence of 28s exposures resulting in 104 frames. 17 frames of this sequence were taken during secondary eclipse and were discarded. The final sequence considered here consisted of 84 frames covering orbital phases $0.533 < \phi < 0.583$. Median SNR's for each observation, as well as the humidity across the entire sequence for each night are shown in Supplementary Fig. 1.

The raw data were calibrated and 1D spectra were extracted per frame by the IGRINS facility team using version 2 of the IGRINS Pipeline Package (PLP, refs. 52,53). We then used the publicly available `cubify` pipeline (https://github.com/petercbsmith/cubify) to process the PLP output into data cuboids of shape $N_{orders} \times N_{frames} \times N_{pixels}$, calculate the planet's orbital phase at each frame as well as the barycentric velocity of the observer for a given time of observation. `cubify` also applies a secondary wavelength realignment to each frame, discarding orders at the edge of the H and K band typically heavily contaminated by tellurics (with a median SNR being approximately < 200), as has been done in previous analysis using IGRINS[14,21,25,54]. `cubify` can also perform a variety of detrending methods to data taken at high spectral

resolution, including singular value decomposition, described in the following paragraph.

The dominant features of each spectrum are the quasi-stationary telluric and stellar lines as well as the wavelength-dependent instrument throughput. To detrend the data, we apply a singular value decomposition (SVD[21,22,55]) to each $N_{frames} \times N_{pixels}$ data matrix per order. This process identifies the most representative features in the data (i.e., the most dominant modes) as the first number of right singular vectors in the matrix decomposition. The removal of these vectors from the data removes the contribution of these dominant modes. Left behind is relatively faint planet signal within the residual data matrix. To allow subsequent reproduction of the SVD effects on each tested model[21,33,56], we recompose the data matrix using the same number right singular vectors originally removed to create a noiseless reconstruction of the raw data. For each night, we elect to remove the first six singular vectors, as this value is enough to remove visible telluric features in most of the IGRINS orders.

## Atmospheric modeling and retrieval frameworks

The high spectral resolution model templates used both for the cross-correlation and retrieval analyses were calculated using a GPU-accelerated version of the exoplanet atmospheric forward modeling code CHIMERA[21]. CHIMERA takes as inputs the atmospheric temperature-pressure profile and the gas volume mixing ratio (VMR) profiles and outputs a high resolution (R = 250,000) thermal emission spectrum given the relevant absorption cross-sections/opacities.

The construction of the temperature-pressure grid utilized for our analysis is similar to the one presented originally in ref. 14. Briefly, the shape of the profile is described by 6 pressure 'nodes', one at the bottom and top of the atmosphere and 4 additional points that can take on any pressure value between the prior ranges listed in Table 1. Each of the pressure nodes has an associated temperature, with prior ranges also listed in Table 1. These nodes define a set of 'control points' within the pressure-temperature space, and it is from these control points that our final grid of temperatures is calculated via the interpolation of these 6 temperature values onto a finer pressure grid using a Bézier Spline. This parametrization of our P-T profile allows for minimal assumptions to be made on its shape, providing flexibility on the lapse rate of the profile while also allowing for thermal inversions in the upper atmosphere, an expected feature of ultra-hot Jupiters due to the increased opacity of optical absorbers[57,58].

To calculate pressure-dependent gas VMR profiles, we pass a given P-T profile and a vector of elemental abundances through the equilibrium chemistry code FASTCHEM[59]. FASTCHEM has routinely been used in the investigation of exoplanet and brown dwarf atmospheres, using instruments from both ground and space[60–62].

We include the opacity of several species with spectral features in the H and K bands that are expected to be present at WASP-189 b's temperatures. This includes the volatile bearing molecules $H_2O$, CO, and OH as well as the following refractory species: Fe I, Mg I, Si I, Ti I, Ca I, V I. We also include the continuum opacity sources from $H_2$-$H_2$ and $H_2$-He collisional induced absorption as well as $H^-$ bound-free and H-$e^-$ free-free absorption. The sources of our opacity line-lists can be found in Supplementary Note 1. High resolution cross-sections are generated over a grid of temperatures and pressures covering several orders of magnitude in atmospheric pressure from $10^2$ to $10^{-6}$ bar and 2000 to 4000 Kelvin (temperatures past 4000K use the final value). Cross-sections for $H_2O$, CO, and OH are generated with the HELIOS-K tool[63] at 0.001 cm$^{-1}$ resolution and a line wing cutoff of 100 cm$^{-1}$ using the information provided in the exomol param files. Atomic cross-sections (over the same grid of pressure and temperature) are generated using a custom routine that considers both natural and pressure (via van-der Walls) broadening. Cross-sections are then interpolated (not averaged or binned) down to a constant R = 250,000 for use within the CHIMERA radiative transfer routines.

The resultant planet thermal emission spectrum is converted to planet-star contrast by dividing the planet spectrum by a PHOENIX stellar model[64] (Teff = 8000 [K], logg = 4.06 [cgs],[19]) interpolated to the IGRINS wavelength grid. This stellar model is further smoothed via a Gaussian filter and finally convolved with a broadening kernel to imitate the line broadening from the IGRINS instrument profile. (The Full-Width at Half Maximum of this broadening kernel is given by the ratio of the resolving power of the model spectrum and that of the instrument, which we assume is constant throughout the H and K bands.) The planet-star contrast is then scaled by the planet-star area ratio. Each planet model spectrum is also convolved with an equatorial rotation kernel assuming tidally locked, solid body rotation. (We do not apply additional rotational broadening, such as those originating from 3D effects, as the effects on the line shape would have minimal impact at the resolving power of IGRINS, e.g., the measured excess rotation speed of 1.35 km/s found in ref. 31).

To account for any stretching or shifting of the underlying planet signal during the detrending procedure[33,56], we apply a final processing step to each planet model spectrum. Before each data-model comparison (either via cross-correlation or likelihood evaluation during the retrieval process), we Doppler shift the model at each phase based on the adopted values for $K_P$ and $dV_{sys}$ and inject it into a noise reduced reconstruction of the data using only the first 6 singular vectors of a given $N_{frame} \times N_{pixel}$ data matrix. We then perform a SVD again on this model-injected matrix to reproduce any alterations to the true planet signal. This last step is crucial to ensure that we apply the same detrending procedure on both the data and the model spectrum. The structure of the atmospheric model is prescribed by an 18 dimensional state vector – 8 elemental abundances along with 10 dimensions from the P-T parameterization. Within our retrieval scheme we also include the planet radial velocity semi-amplitude, $K_P$, and deviation from the expected system velocity, $dV_{sys}$, for each observational sequence as nuisance parameters, resulting in a total 22 free parameters.

To sample the posterior distribution, we use a affine invariant ensemble sampler through the python emcee[35] package, initializing our sampler with 88 walkers. Our likelihood function is described by the CCF-to-likelihood mapping framework from[33], which has been utilized in several previous HRCCS studies[14,22,54], and validated against other likelihood framework developed such as[34] and traditional chi-squared-based likelihood approach used on data from the James Webb Space Telescope[25]. The prior ranges assumed for all parameters are detailed in Table 1. We ran the sampler for 10,000 iterations monitoring the autocorrelation length scale, parameter medians, and variance. Running for up to 50,000 iterations changed the nominal metrics by very little. For our analysis, we select the last 1000 chains, and are left with a set of 88,000 total samples. The resultant posterior probability distribution is summarized in Supplementary Figs. 3–5. The the median P-T profile from 1000 random draws from the posterior probability distribution is shown in Supplementary Fig. 6.

## Gas detection via cross-correlation

We calculate 2D cross-correlation function (CCF) maps (presented in Fig. 3) as is standard in the high resolution exoplanet spectroscopy literature[24,55]. This is done by calculating the Pearson correlation coefficient between the post-SVD data and an atmospheric model template spectrum Doppler shifted at each orbital phase/spectrum/frame in accordance with the planet's radial velocity semi-amplitude, $K_P$, and a systematic velocity offset, $dV_{sys}$. We do this along a grid of possible $K_P$ and $dV_{sys}$ values, spanning 200 km s$^{-1}$ along each velocity dimension centered on the literature value, resulting in 2D maps of correlation coefficients. We quantify the planet signal-to-noise ratio (S/N) by subtracting the median and normalizing by the standard deviation of a 3$\sigma$-clipped copy of the CCF map, using the *Astropy*

function `sigma_clipped_stats`. The calculation of the CCF signal-to-noise using this method has been applied in previous studies[14], and is warranted given the large range in $K_P$ and $V_{sys}$ values we explore within the parameter space. The adopted planet detection S/N, i.e., the planet signal detection significance, is the maximum of this normalized CCF map localized around a 25 km/s box centered around the literature planet $K_P$ and $V_{sys}$. The resultant S/N in most of these strong detections is likely an underestimate of the true detection significance, as, given the level of significance of the detections, the noise structure at $K_P$-$V_{sys}$ pairs far from the peak is likely a mix of both noise and aliased signal.

To search for individual gases, we use the same P-T and gas VMR profiles used to calculate the Fiducial model, but set the abundances of all gases to zero except the specific gas of interest and continuum opacity sources. We then repeat the process and recalculate a high resolution model spectrum and CCF S/N map. The individual detections of Fe I, CO, OH, Mg I, Si I, $H_2O$ are shown in 3. This marks the second simultaneous detection of both Fe I and Si I in the atmosphere of UHJ in the infrared[65]. The presence of OH is indicative of the thermal dissociation of $H_2O$, which may suggest why do not detect water as strongly as other molecules less susceptible to thermal dissociation at these temperatures such as CO. Additionally, the detections of Fe I and Mg I in emission are consistent with previous observations of WASP-189b in transmission[61].

For CCF maps containing both multiple and single species templates, we consistently find a peak for the cross-correlation function at a dVsys of approximately 4 km/s and a Kp at approximately 193 km/s. This is consistent with our retrieval values where we obtain a dKp (delta Kp value) of $-11.84^{+0.92}_{-0.83}$ and $-11.71^{+1.77}_{-1.75}$, and dVsys values of $5.25^{+0.49}_{-0.45}$ and $2.95^{+0.70}_{-0.57}$ for each night respectively, (see Supplementary Fig. 5). Here, Kp was assumed to have the literature value of 201 ± 4 km/s, calculated using the semi-major axis measurement from[18] and the orbital period from[66]. However, values are in agreement with previous high-resolution thermal emission studies of WASP-189b[31,67] who measure Keplerian velocities of $193.54^{+0.54}_{-0.53}$ km/s and $193.40^{+2.4}_{-2.5}$ km/s.[31] also measure a delta $V_{sys}$ offset 4.7 ± 0.8 km/s. While all three measurements of Kp are offset from the assumed literature value, they are well within the uncertainties of the Kp value quoted in ref. 67, who calculate a literature Kp value using both the period and semi-major axis from[66], calculated at $197^{+15}_{-16}$ km/s.

Searches for individual signal of Ti I, Ca I, V I resulted weak-to-non detections. While there are some lines for these species in the H and K bands, these species have fewer lines in the near-infrared compared to the species which we detect confidently. When we searched for these species using the likelihood formalism of[33] (the same as used in our main retrieval algorithm), as opposed to the traditional Pearson correlation coefficient, these species are seen with tentative significances. This is expected, as the likelihood function is more sensitive to the amplitudes and line-shapes than the traditional correlation coefficient, an effect which has been shown in previous HRCCS gas detections[14]. We show the CCF and log-likelihood detection maps for Ti I, Ca I, V I in the top and bottom rows of Supplementary Fig. 2.

## Data availability
This work is based on observations made with the Gemini South Telescope. The raw data products are available within the Gemini Observatory Archive [https://archive.gemini.edu/searchform] under program IDs GS-2021B-Q-113 and GS-2023A-Q-231. The data generated and analyzed in this study, including reduced data cubes, model spectra, files used to generate cross correlation maps, and the main retrieval outputs have been deposited in following Zenodo repository via https://doi.org/10.5281/zenodo.18462071. Stellar abundances quoted in this analysis are from refs. 18, 41 and 68. Solar abundances quoted in this analysis are from[40]. Source data are provided with this paper.

## Code availability
The CHIMERA code used in this analysis is based upon that presented in ref. 21, and can be obtained as described in ref. 21. The data products used in the analysis were reduced using the `cubify` package: https://zenodo.org/records/14194202. The code also made use of the publicly available `fastchem` tool[59]: https://github.com/NewStrangeWorlds/FastChem. This analysis also made use of the nested-sampling package pymultinest: https://johannesbuchner.github.io/PyMultiNest/, joblib loop parallelization package: https://joblib.readthedocs.io/en/latest/, and corner.py: https://corner.readthedocs.io/en/latest/.

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

## Acknowledgements

J.A.S., M.R.L., S.K.K., and J.L.B. acknowledge support from NSF grant AST-2307177/8. P.C.B.S. acknowledges support provided by NASA through the NASA FINESST grant 80NSSC22K1598. L.W. and M.W.M. acknowledge support through the 51 Pegasi b Fellowship awarded by the Heising-Simons Foundation. The results reported herein benefited from collaborations and/or information exchange within NASA's Nexus for Exoplanet System Science (NExSS) research coordination network sponsored by NASA's Science Mission Directorate, grant 80NSSC23K1356, PI Steve Desch. This material is based upon work supported by the National Science Foundation under grant MSIP-1836008. We acknowledge Research Computing at Arizona State University for providing high-performance computing and storage resources that have significantly contributed to the research results reported within this manuscript. This work used the Immersion Grating Infrared Spectrometer (IGRINS) that was developed under a collaboration between the University of Texas at Austin and the Korea Astronomy and Space Science Institute (KASI) with the financial support of the Mount Cuba Astronomical Foundation, of the US National Science Foundation under grants AST-1229522 and AST-1702267, of the McDonald Observatory of the University of Texas at Austin, of the Korean GMT Project of KASI, and Gemini Observatory. This program is based on observations obtained at the international Gemini Observatory, a program of NSF's NOIRLab, which is managed by the Association of Universities for Research in Astronomy (AURA) under a cooperative agreement with the National Science Foundation on behalf of the Gemini Observatory partnership: the National Science Foundation (United States), National Research Council (Canada), Agencia Nacional de Investigación y Desarrollo (Chile), Ministerio de Ciencia, Tecnología e Innovación (Argentina), Ministério da Ciência, Tecnologia, Inovaç μes e Comunicaç μes (Brazil), and Korea Astronomy and Space Science Institute (Republic of Korea). J.A.S. and M.R.L. would especially like to thank the anonymous queue observers who successfully completed the observing programs this work is based on. We would like to thank Monika Lendl for useful discussion on stellar abundance analyses.

## Author contributions

J.A.S. performed the analysis, wrote the manuscript, and prepared one of the proposals that resulted in the post-eclipse dataset. P.C.B.S. developed code, contributed text to the manuscript, provided scientific guidance, and reviewed the paper. K.K. performed additional retrieval analyses and integrated the FastChem code into the CHIMERA retrieval pipeline. L.W. provided scientific input, guided J.A.S. throughout the writing process, and advised on manuscript focus. M.R.L. conceived the paper, wrote the original Gemini L.L.P. proposal from which the pre-eclipse dataset originates, edited and revised the manuscript, and supervised J.A.S. throughout the project. S.P. provided code and spectral comparisons that helped identify a bug in the atomic opacities. S.D., P.Y., and J.P. provided comments on multiple drafts and scientific guidance on planetary formation and composition. J.B., M.B., D.J., and G.N.M. contributed to the development and operation of IGRINS and its associated data pipeline. M.W.M. and A.B.S. contributed to data analysis, retrieval tests, and manuscript comments. V.P., L.P., L.v.S., and J.P.W. provided theoretical context on atmospheric chemistry and manuscript feedback. All authors read and approved the final version of the manuscript and contributed to the discussion of results.

## Competing interests

The authors declare no competing interests.
