## [Transparent Peer Review file · Nature Communications]

A Stellar Magnesium to Silicon ratio in the atmosphere of an exoplanet

Corresponding Author: Mr Jorge Sanchez

Version 0:

Reviewer comments:

Reviewer #1

(Remarks to the Author)

I acknowledge that the authors have addressed most of the points raised during the first round of revisions. The text and figures are now clearer, and the manuscript is more readable and of higher quality overall. As I mentioned in my first referee report, I still believe that the findings presented in this paper are relevant to the exoplanet community.

However, one important concern seems to remain insufficiently addressed: It is still unclear to me how measurements of ultra-hot Jupiter atmospheric elemental ratios can provide significant insight into the nature of rocky exoplanets. At most, these measurements appear to support the assumption that stellar refractory elemental ratios are a reasonable basis for modeling the interiors of rocky planets. Beyond that, however, I do not see a clear, direct constraint on the compositions of rocky exoplanets; therefore, the connection remains speculative in my eyes.

This concern is particularly evident in lines 138–141, where it is claimed that the work on WASP-189b helps to bridge the gap between observations and the use of stellar refractory ratios as a proxy for rocky exoplanets. It appears unclear in what sense this gap is being bridged? I would therefore encourage the authors to either clarify the specific mechanism by which this study informs about rocky exoplanet composition or present the claim more cautiously rather than describing it as a key finding.

That said, I find the results reported here to be highly relevant, particularly to our understanding of WASP-189b and similar close-in giant exoplanets. The manuscript communicates important science and, with the above clarification, will make a strong contribution to the field.

Reviewer #2

(Remarks to the Author)

The authors present high-resolution IGRINS observations of dayside emission for the Ultra-hot Jupiter WASP-189b, which are used to constrain the abundances of key refractory and volatile species (namely O, C, Fe, Mg, Si). WASP-189b joins a small sample of planets that have both measured abundances of volatiles and refractories, which has been proposed as an important diagnostic for constraining planet formation processes and planet formation location in the disk. Measuring both the refractory and volatile abundances from the same spectra is also very impressive and demonstrates the real power of the wide infrared wavelength coverage that is somewhat unique to IGRINS.

In a previous review of this paper, I brought up concern over the language emphasizing the importance of the Mg/Si ratio as motivation for the paper and as the main focus of the text due to the expectation that Mg, Si, Fe will all be in solid form throughout the disk and are therefore all expected to act very similarly in how they are inherited into planets. That being said, I do think that it is important to check/ensure that abundances are indeed as expected, and so confirming that the Mg/Si, Si/Fe, Mg/Fe ratios are solar is worth doing and an important result of the paper, and I appreciate how the authors have amended the abstract to reflect that the result is a test of the assumption of solar refractory abundances.

Remaining Major Concerns:

1. While the abstract has been amended and I think the focus on confirming the assumption of solar Mg/Si is more appropriate, I still think that the discussion about terrestrial planet mineralogy is an extrapolation (this includes, the last

sentence of the abstract as well as the whole paragraph between lines 130 and 141). The Mg, Si, and Fe abundances in the envelope of a giant planet are likely accreted in a very different way as to how material builds up in a planetary core. More citations and justification about the validity of extrapolating abundances from giant exoplanet atmospheres/envelopes to terrestrial planet interiors needs to be provided in order for the authors to discuss a hypothetical terrestrial planet with an upper mantle akin to that of the Earth.

2. I did not see that the authors added any text about the use of equilibrium chemistry, and so I will add the same concern again in this report. Can the authors justify why they believe equilibrium chemistry is a good assumption for this dataset and why they did not pursue the "free retrieval" approach? Disequilibrium processes have been shown to be present on the daysides of ultra-hot Jupiters (e.g., Baxter+2021, Baeyens+2024, Saha & Jenkins 2025, etc.) and so the authors should either show that their results would be unchanged by the chemistry prescription, or justify why the assumption of equilibrium chemistry is valid for this case.

Finally, I think that one of the main utilities coming out of this paper is that if the Mg/Si, Si/Fe, Mg/Fe ratios are all about solar, this means that these species can be used interchangeably in estimating the "rock" abundance as it relates to measuring the "rock-to-volatile" enhancement. This rock to volatile abundance has emerged as a powerful tool for understanding giant planet formation and evolution, and if the overall rock abundance can be accurately measured by just constraining a single one of these refractory species that would make these measurements significantly less time intensive and also allow observers to go after the most observationally accessible refractory species. The authors do not discuss this point at all in the manuscript, and in my opinion this would strengthen the applicability of the paper.

Version 1:

Reviewer comments:

Reviewer #1

(Remarks to the Author)

I thank the authors for their response and for revising the paper in accordance with my previous comments and concerns. I particularly appreciate the first paragraph of their reply, which clearly summarizes the key finding of the paper, namely "...that refractory ratios in giant-planet atmospheres provide empirical validation of the stellar-proxy assumption that underpins planetary composition models".

I have no further comments; from my side the paper is ready for publication.

Reviewer #2

(Remarks to the Author)

I think the authors have nicely reworked the text in response to my concerns (as well as the same concerns from the other referee), and I think that the discussion of how the derived abundance ratios in WASP-189b relate to terrestrial planet formation are now appropriate.

I also appreciated the authors thorough investigation into the question of equilibrium chemistry vs. free retrievals. They have added the many references they discuss in the response to my report to the text, but I do think many readers would find a single sentence or two explanation (and not just the list of citations) useful to quickly understand why equilibrium chemistry is the preferred choice over free retrievals without having to dig through many references. The authors provided this in the response and so I would think it would be easy to add a summarizing sentence to the text as well.

At this point I am happy to see the paper published and I think the authors have sufficiently addressed my concerns.

We thank the editor and referees for their further review of this manuscript. This process has substantially improved both the clarity and the impact of the work. In this revision, we have reworked the discussion of the rocky-planet connection, expanded our justification for the use of equilibrium chemistry, and added key references clarifying how giant-planet atmospheres trace the stellar refractory ratios. Collectively, these changes resolve the conceptual concerns raised in the previous round and strengthen the manuscript's central conclusion that refractory ratios in giant-planet atmospheres provide empirical validation of the stellar-proxy assumption that underpins planetary composition models. As with the previous report, changes made to the manuscript have been added as blue text.

Reviewer #1 (Remarks to the Author):

I acknowledge that the authors have addressed most of the points raised during the first round of revisions. The text and figures are now clearer, and the manuscript is more readable and of higher quality overall. As I mentioned in my first referee report, I still believe that the findings presented in this paper are relevant to the exoplanet community.

However, one important concern seems to remain insufficiently addressed: It is still unclear to me how measurements of ultra-hot Jupiter atmospheric elemental ratios can provide significant insight into the nature of rocky exoplanets. At most, these measurements appear to support the assumption that stellar refractory elemental ratios are a reasonable basis for modeling the interiors of rocky planets. Beyond that, however, I do not see a clear, direct constraint on the compositions of rocky exoplanets; therefore, the connection remains speculative in my eyes.

This concern is particularly evident in lines 138–141, where it is claimed that the work on WASP-189b helps to bridge the gap between observations and the use of stellar refractory ratios as a proxy for rocky exoplanets. It appears unclear in what sense this gap is being bridged? I would therefore encourage the authors to either clarify the specific mechanism by which this study informs about rocky exoplanet composition or present the claim more cautiously rather than describing it as a key finding.

That said, I find the results reported here to be highly relevant, particularly to our understanding of WASP-189b and similar close-in giant exoplanets. The manuscript communicates important science and, with the above clarification, will make a strong contribution to the field.

We thank the referee for their kind words on the quality of the manuscript for this thoughtful follow-up.

We agree that giant and rocky planets follow distinct formation pathways and have further clarified that our aim is not to infer detailed rocky-planet compositions, but to provide an empirical test of the long-standing assumption that stellar refractory ratios are preserved in planetary material, as the referee correctly points out. This assumption is motivated by the planet formation simulations by, e.g. Thiabaud et al. 2015, Lothringer et al. 2021, and Chachan et al. 2023. Specifically, Thiabaud et al. use planet-formation simulations to predict that Mg/Si and Fe/Si ratios remain nearly stellar for both giant and terrestrial planets formed beyond the ice

line, with only minor deviations for planets that form very close to the star. Our results provide an observational validation of this theoretical expectation.

We have effectively re-worked (also suggested by referee 2) the paragraph between lines 128-140 (now 134-146):

To explore the geochemical context of our measured abundance ratios, we calculate the relative proportions of Mg, Si, and Fe in WASP-189b's atmosphere. Based on the retrieved abundances, we derive a Mg:Si:Fe ratio of approximately 1.2:1.0:0.7, which falls between the typical values found in enstatite and ordinary chondrites \cite{Lodders2003}. If a rocky planet were to form from material with this composition, its upper mantle would likely consist of a mixture of olivine and pyroxene, similar to Earth's \cite{Dorn2017,Untertorn2017}. While we do not claim that giant planet atmospheres and rocky planet interiors evolve identically, both draw from the same disk reservoir of refractory solids \cite{Thiabaud2015}. The atmospheres of gas giants—particularly those with minimal condensation or cold-trapping (like WASP-189b, see Methods)—are expected to retain the elemental ratios of the accreted refractory species, thereby serving as observable tracers of the disk's bulk refractory composition \cite{othringer21,Chachan2023}. The consistency of WASP-189b's atmospheric Mg/Si, Mg/Fe, and Si/Fe ratios with the host star supports the assumption—widely used in terrestrial planet interior modeling \cite{Dorn2017,Untertorn2017}—that stellar refractory abundances trace the bulk composition of solids in the protoplanetary disk \cite{Thiabaud2015,Adibekyan2021,Chachan2023,Spaargaren2020}. In this sense, measurements of these ratios in ultra-hot Jupiters offer an observational anchor for understanding the link between planet and host star composition, and ultimately the refractory building blocks in planetary systems.

We also removed the calculations of the accreted masses of Fe, Mg, and Si from the methods section as it is no longer relevant.

Reviewer #2 (Remarks to the Author):

The authors present high-resolution IGRINS observations of dayside emission for the Ultra-hot Jupiter WASP-189b, which are used to constrain the abundances of key refractory and volatile species (namely O, C, Fe, Mg, Si). WASP-189b joins a small sample of planets that have both measured abundances of volatiles and refractories, which has been proposed as an important diagnostic for constraining planet formation processes and planet formation location in the disk. Measuring both the refractory and volatile abundances from the same spectra is also very impressive and demonstrates the real power of the wide infrared wavelength coverage that is somewhat unique to IGRINS.

In a previous review of this paper, I brought up concern over the language emphasizing the importance of the Mg/Si ratio as motivation for the paper and as the main focus of the text due to the expectation that Mg, Si, Fe will all be in solid form throughout the disk and are therefore all expected to act very similarly in how they are inherited into planets. That being said, I do

think that it is important to check/ensure that abundances are indeed as expected, and so confirming that the Mg/Si, Si/Fe, Mg/Fe ratios are solar is worth doing and an important result of the paper, and I appreciate how the authors have amended the abstract to reflect that the result is a test of the assumption of solar refractory abundances.

We thank the referee for acknowledging the improved manuscript and the significance of the fundamental measurement we have made.

Remaining Major Concerns:

1. While the abstract has been amended and I think the focus on confirming the assumption of solar Mg/Si is more appropriate, I still think that the discussion about terrestrial planet mineralogy is an extrapolation (this includes, the last sentence of the abstract as well as the whole paragraph between lines 130 and 141). **The Mg, Si, and Fe abundances in the envelope of a giant planet are likely accreted in a very different way as to how material builds up in a planetary core.** More citations and justification about the validity of extrapolating abundances from giant exoplanet atmospheres/envelopes to terrestrial planet interiors needs to be provided in order for the authors to discuss a hypothetical terrestrial planet with an upper mantle akin to that of the Earth.

We thank the referee for this thoughtful comment. This comment is similar to Referee 1 above, and thus we reproduce our response here for convenience.

We agree that giant and rocky planets follow distinct formation pathways and have further clarified that our aim is not to infer detailed rocky-planet compositions, but to provide an empirical test of the long-standing assumption that stellar refractory ratios are preserved in planetary material, as the referee correctly points out. This assumption is motivated by the planet formation simulations by, e.g. Thiabaud et al. 2015, Lothringer et al. 2021, and Chachan et al. 2023. Specifically, Thiabaud et al. use planet-formation simulations to predict that Mg/Si and Fe/Si ratios remain nearly stellar for both giant and terrestrial planets formed beyond the ice line, with only minor deviations for planets that form very close to the star. Our results provide an observational validation of this theoretical expectation.

We have effectively re-worked (also suggested by referee 2) the paragraph between lines 128-140 (now 134-146):

To explore the geochemical context of our measured abundance ratios, we calculate the relative proportions of Mg, Si, and Fe in WASP-189b's atmosphere. Based on the retrieved abundances, we derive a Mg:Si:Fe ratio of approximately 1.2:1.0:0.7, which falls between the typical values found in enstatite and ordinary chondrites \cite{Lodders2003}. If a rocky planet were to form from material with this composition, its upper mantle would likely consist of a mixture of olivine and pyroxene, similar to Earth's \cite{Dorn2017,Untertorn2017}. While we do not claim that giant planet atmospheres and rocky planet interiors evolve identically, both draw

from the same disk reservoir of refractory solids \cite{Thiabaud2015}. The atmospheres of gas giants—particularly those with minimal condensation or cold-trapping (like WASP-189b, see Methods)—are expected to retain the elemental ratios of the accreted refractory species, thereby serving as observable tracers of the disk’s bulk refractory composition \cite{lothringer21,Chachan2023}. The consistency of WASP-189b’s atmospheric Mg/Si, Mg/Fe, and Si/Fe ratios with the host star supports the assumption—widely used in terrestrial planet interior modeling \cite{Dorn2017,Unterborn2017}—that stellar refractory abundances trace the bulk composition of solids in the protoplanetary disk \cite{Thiabaud2015,Adibekyan2021,Chachan2023,Spaargaren2020}. In this sense, measurements of these ratios in ultra-hot Jupiters offer an observational anchor for understanding the link between planet and host star composition, and ultimately the refractory building blocks in planetary systems.

We also removed the calculations of the accreted masses of Fe, Mg, and Si from the methods section as it is no longer relevant.

Furthermore, we modified the last sentence of the abstract to read :

“More broadly, this work demonstrates that atmospheres can preserve geochemical fingerprints of formation and supports the use of stellar proxies in interpreting the bulk compositions of exoplanets”

Finally, we removed the calculations of the accreted masses of Fe, Mg, and Si from the methods section as it is no longer relevant.

2. I did not see that the authors added any text about the use of equilibrium chemistry, and so I will add the same concern again in this report. Can the authors justify why they believe equilibrium chemistry is a good assumption for this dataset and why they did not pursue the “free retrieval” approach? Disequilibrium processes have been shown to be present on the daysides of ultra-hot Jupiters (e.g., Baxter+2021, Baeyens+2024, Saha & Jenkins 2025, etc.) and so the authors should either show that their results would be unchanged by the chemistry prescription, or justify why the assumption of equilibrium chemistry is valid for this case.’

We thank the referee for pointing this out. We have now explicitly justified our use of equilibrium chemistry in the main text (line 82). In summary, at WASP-189b’s dayside temperature ($\sim > 3000$ K), thermochemical timescales are short, and multiple studies confirm equilibrium is valid for major species at observable pressures.

We provided a lengthy response as well as supporting references in the main text in the previous round of revisions, and have reproduced and expanded here.

Free retrievals, while we agree are informative for select cases, have numerous limitations when applied in the UHJ regime. In particular, a standard free retrieval approach assumes constant-with-altitude

abundance profiles, which has been shown to fail to capture the important effects of dissociation (e.g., Parmentier et al. 2018, Brogi et al. 2023). Parameterized profiles can be used, however, these are almost exclusively based upon the thermochemical equilibrium based parameterization from Parmentier et al. 2018. Effectively, a free retrieval assuming this parameterization is largely assuming equilibrium chemistry. Weiner Mansfield et al. 2024 took a more flexible approach with the water, CO, and OH profiles, applied to the UHJ WASP-76b IGRINS transmission spectrum, and found agreement with the equilibrium chemistry assumption.

These intrinsic challenges have been overcome by using chemical equilibrium models. Their use has been validated (as discussed above) and justified theoretically. For instance, it is a widely held assumption that the chemistry of the primary C and O reservoirs in planets in this temperature regime (>2000 K) is well described by thermochemical equilibrium at all but the lowest of pressures ($<\sim 10^{-5}$ bars) owing to the relatively short kinetic timescales compared to mixing and photodissociation processes. This assumption is supported by the literature (e.g., Koppurapu et al. 2011, Moses et al. 2013, Line & Yung 2013, Coulombe et al. 2023). Thermochemical equilibrium retrievals have been applied as standard practice on both low and hi-res observations of UHJs (e.g. Kreidberg et al. 2015, Pelletier et al. 2022, Brogi et al. 2023, Smith et al. 2024, Pelletier et al. 2025, and numerous others).

Additionally, in the UHJ regime, numerous elements have some fraction in the ionized state (e.g., Fe II, Mg II, etc.) or molecules in a dissociated state (e.g., water to OH and O). Few, if any, of the ion lines persist in the IR wavelengths (mostly in the blue optical), nor any lines from atomic O. Therefore, attempting to constrain the entire elemental inventory of a single element (e.g., Fe, Mg, O) within a free retrieval framework would struggle owing to missing a fraction of that species in the ionized (or dissociated, in the case of O) state. Equilibrium chemistry naturally accounts for these effects. That is, given the elemental abundance and the P-T conditions, equilibrium dictates which fraction of that element is in the molecular vs. neutral atomic vs. ionized atomic state. The lines we are probing arise from the neutral state transitions (e.g. Fe I, Mg I, Si I, etc.). Therefore, probing these neutral state lines, through the equilibrium chemistry, we can ascertain the intrinsic elemental (e.g., Fe/H) abundance, despite not having access to the ionized state transitions. Brogi et al. 2023 elucidate (their sections 5.1 and 5.2 and Figures 3 and 8) this issue in the context of the carbon-to-oxygen ratios derived from free retrievals. Specifically, a large fraction of the total oxygen inventory resides in atomic O, which is not probed by infrared wavelengths. Thus measuring only the water, CO, (and potentially OH) abundances would in fact miss about $\sim 40\%$ of the oxygen inventory and arrive at an artificially high C/O. The solution to this issue was to apply an equilibrium-chemistry based retrieval.

Finally, with free retrievals it's far too easy to "overfit" or bias the results by choosing too limited of a set of molecules (e.g., see Welbanks et al. 2025, accepted). Effectively, planets similar to wasp-189 have been well described by thermochemical equilibrium.

The disequilibrium chemistry references suggested here are interesting, but upon closer inspection they do not convincingly demonstrate the significance of disequilibrium chemistry in UHJ's. Baxter et al. 2021 (their Figure 6) shows that disequilibrium chemistry has virtually no effect on planets hotter than ~ 1500 K for all but the lowest gravity model (wasp-189 falls in between their 1500 and 5000 cm/s^2 scenarios). Furthermore, their investigation did not include thermal inversions in the UHJ's (they used analytic

Guillot 2010 type profiles without a large visible opacity parameter), which vastly underestimates the temperatures in the mid to upper atmosphere—hotter tends towards equilibrium. Baeynes et al. 2024 is the closest to demonstrating the importance of disequilibrium processes, however, only for pressures generally below $<1e-4$ bars for molecules in question—largely outside of our sensitivity limits. Finally, Saha & Jenkins is a submitted manuscript, not yet accepted, so it remains to be seen how this result holds. Their results are based upon elevated abundances of carbon species, in particular C₂H₂. However, this result is not robust against the different data reductions used. That manuscript only compares a free retrieval to an equilibrium retrieval and argues that because the Bayes factor is higher for the free retrieval, that disequilibrium chemistry must be occurring. This approach/argument is questionable in light of recent literature (e.g., Welbanks et al. 2025, accepted) whereby free retrievals are subject to overfitting and can provide overconfidence in Bayes factors dependent upon the selected molecules to include in the retrieval. That manuscript never attempts to explain their retrieved free retrieval abundances using any kind of disequilibrium chemistry models, so it may in fact be true, that no physically motivated model can explain their free retrieval abundances.

In short, we argue that the validity of the assumption of equilibrium chemistry applied as we have done to WASP-189b has already been demonstrated/validated in the literature. **We have added the relevant references supporting this assumption in the main text line 82.**

Finally, I think that one of the main utilities coming out of this paper is that if the Mg/Si, Si/Fe, Mg/Fe ratios are all about solar, this means that these species can be used interchangeably in estimating the "rock" abundance as it relates to measuring the "rock-to-volatile" enhancement. This rock to volatile abundance has emerged as a powerful tool for understanding giant planet formation and evolution, and if the overall rock abundance can be accurately measured by just constraining a single one of these refractory species that would make these measurements significantly less time intensive and also allow observers to go after the most observationally accessible refractory species. The authors do not discuss this point at all in the manuscript, and in my opinion this would strengthen the applicability of the paper.

We appreciate this suggestion and have added a short discussion at lines 115-120 emphasizing the observational and modeling utility of this finding:

"The consistency of Mg, Si, and Fe with stellar values indicates that these elements behave interchangeably as refractory species, as theoretically predicted \cite{Chachan2023,Lothringer2021}. Consequently, constraining any one of these species may suffice to estimate a planet's total refractory enrichment and [R/V] ratio (provided that species is not subject to condensation), simplifying future comparative studies of disk composition and formation.

We thank Referee #2 for their final comment, which states:

I also appreciated the authors thorough investigation into the question of equilibrium chemistry vs. free retrievals. They have added the many references they discuss in the response to my report to the text, but I do think many readers would find a single sentence or two explanation (and not just the list of citations) useful to quickly understand why equilibrium chemistry is the preferred choice over free retrievals without having to dig through many references. The authors provided this in the response and so I would think it would be easy to add a summarizing sentence to the text as well.

We address this concern by adding the following text in the revised manuscript near lines 116 - 120:

The atmospheric forward model assumes \cite{Lesjak2024, pelletier2025} thermochemical equilibrium \cite{fastchem} for the molecular and atomic gases in WASP-189b **given the elemental abundance ratios and the temperature-pressure profile.** **This parameterization is preferred for ultra-hot Jupiters as the high temperatures result in kinetic timescales that are short compared to disequilibrium processes (photochemistry, transport) timescales, resulting in equilibrium gas abundances throughout the atmosphere \cite{Kopparapu2012, Moses2013a, Lothringer2018}.**